# Local Redundancy: An Information-Theoretic Measure of Plasticity from Synthetic Memorization

Jiaxuan Cheng [1]

## Abstract

Plasticity—a neural network's ability to adapt to new tasks—is critical for continual and transfer learning. Existing measures, such as effective rank, dead neuron fraction, and weight norm, lack theoretical grounding and correlate poorly with performance on new tasks. We introduce *local redundancy*, an information-theoretic measure derived from universal compression theory. We define local redundancy as the worst-case redundancy of a local model family—parameters in an infinitesimal neighborhood along gradient directions—and show this is a principled measure of plasticity. Although local redundancy is intractable to compute exactly, we prove that the expected squared gradient norm on a synthetic memorization task provides an efficiently computable lower bound. Experiments on continual image classification and time series transfer learning demonstrate that local redundancy predicts downstream performance better than existing measures and enables pretraining checkpoint selection where validation loss plateaus.

## 1. Introduction

Neural networks trained over long horizons or across multiple tasks often lose the ability to learn effectively, a phenomenon termed *loss of plasticity* (Lyle et al., 2023; Dohare et al., 2024). Modern training pipelines increasingly involve multiple adaptation phases such as fine-tuning, alignment, and continual updates, where plasticity loss in early stages compromises learning in later ones. Understanding and measuring plasticity is therefore central to training neural networks that learn continuously.

Prior work has identified several properties of neural networks that change as plasticity degrades: lower effective rank of representations (Kumar et al., 2021), higher dormant neuron ratio (Sokar et al., 2023), larger weight norms (Lyle et al., 2023), and greater distance from initialization. These quantities are often used as proxies for plasticity, but each captures only one facet of network degradation without a unifying principle. In practice, we find they correlate poorly with future task performance.

We introduce *local redundancy*, a measure of plasticity derived from universal compression theory. The redundancy of a model class measures the excess loss incurred by any predictor that must handle all distributions in the class (Shtarkov, 1987; Rissanen, 1984); we define local redundancy as the worst-case redundancy of an infinitesimal neighborhood of parameters along gradient directions. Classically (and in singular model families appearing in deep learning), worst-case redundancy is asymptotically equal in leading term to the information radius of the model class—the diversity of distributions the model can express (Haussler & Opper, 1997; Watanabe, 2009). This measures the capacity of the network to fit distributions different from its training data (Figure 1).

We prove that the expected squared gradient norm on synthetic memorization data provides a lower bound on local redundancy (Theorem 3.4). The proof connects memorization gain—the reduction in loss when fitting random labels—to worst-case redundancy via the Shtarkov sum. The memorization dataset consists of synthetic images paired with randomly sampled labels, with inputs designed to saturate the network's memorization capacity. We show that lower bounding redundancy via memorization yields smooth training curves, and thus the synthetic gradient norm acts as both a lower bound and a proxy to estimate local redundancy.

We evaluate local redundancy in two settings and find it outperforms existing proxies. In continual image classification over thousands of sequential tasks, local redundancy correlates higher with future task accuracy than effective rank, dormant neuron ratio, and weight norm, after accounting for task number. In time series transfer learning, selecting pretrain checkpoints with maximum local redundancy achieves better adaptation performance than selecting the checkpoint with lowest validation loss (Table 3), something

[1]Massachusetts Institute of Technology, Cambridge, MA, USA. Correspondence to: Jiaxuan Cheng <jasonc75@mit.edu>.

*Proceedings of the 43rd International Conference on Machine Learning*, Seoul, South Korea. PMLR 306, 2026. Copyright 2026 by the author(s).

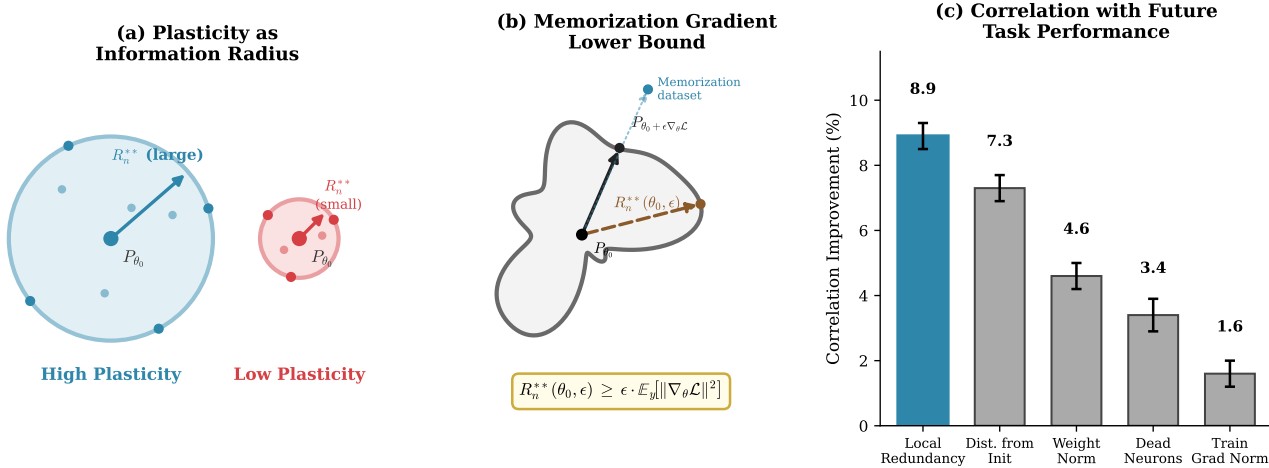

*Figure 1.* **Local redundancy: a principled, computable measure of plasticity.** (a) Redundancy measures the information radius of a model class: a plastic network can reach many distributions (large radius); a rigid network cannot (small radius). (b) The expected squared gradient norm on synthetic memorization data lower-bounds local redundancy (Theorem 3.4), requiring only a single backward pass to compute. (c) Local redundancy predicts downstream performance better than existing plasticity heuristics in continual learning; see Section 4 for transfer learning.

not achievable with other plasticity metrics.

Our contributions are as follows:

- We define local redundancy, an information-theoretic measure of plasticity based on the worst-case redundancy of a network's local parameter neighborhood.

- We prove that the expected squared gradient norm on synthetic memorization data lower-bounds local redundancy, computable in a single backward pass.

- We show that local redundancy predicts future task performance better than existing proxies and enables checkpoint selection when validation loss plateaus.

## 2. Related Work

**Information theory and universal compression.** Our framework builds on information-theoretic approaches to learning, specifically the theory of universal compression (Rissanen, 1984; Barron et al., 1998). The minimax redundancy characterizes the unavoidable cost of coding without knowledge of the source (Davisson, 1973; Shtarkov, 1987), is equivalent to the capacity of the channel induced by the model family (Haussler & Opper, 1997; Merhav & Feder, 1998). For classical regular models, redundancy scales as $\frac{d}{2} \log n$ where $d$ is the parameter dimension (Rissanen, 1996). For singular models such as neural networks, singular learning theory generalizes this result and replaces the effective dimension with the real log canonical threshold (RLCT) (Watanabe, 2009). This has recently been applied to study neural network learning dynamics (Hoogland

et al., 2024; Chen et al., 2023) and information-theoretic generalization bounds based on mutual information (Xu & Raginsky, 2017; Steinke & Zakynthinou, 2020). We apply ideas from universal compression theory to neural networks, using redundancy as a measure of plasticity.

**Model complexity.** Classical model complexity measures include VC dimension (Vapnik & Chervonenkis, 1971) and Rademacher complexity (Bartlett & Mendelson, 2002), which characterize global worst-case learnability but are difficult to compute for deep networks and insensitive to the current parameter configuration. Dong et al. (2025) recently obtain tight bounds via Jensen-Shannon divergence. Munn & Wei (2025) use MDL-based model selection to choose pretraining checkpoints among pretrain runs with differing hyperparameters or model families. In contrast, local redundancy measures capacity at a specific parameter configuration and enables checkpoint selection within a single run.

**Plasticity and continual learning.** Loss of plasticity, where networks progressively lose the ability to learn new tasks, has been observed across reinforcement learning (Lyle et al., 2023; Dohare et al., 2024; Abbas et al., 2023; Nikishin et al., 2022) and supervised settings (Ash & Adams, 2020; Achille et al., 2019). Proposed causes include rank collapse of representations (Kumar et al., 2021), dormant neurons (Sokar et al., 2023), saturated activations, growing weight norms (Lyle et al., 2023), and drift from initialization. The Fisher information matrix and Hessian have also been used to measure plasticity through the curvature of the loss landscape (Kirkpatrick et al., 2017; Martens, 2020; LeCun et al., 1989). These measure local curvature with respect to the

current training distribution; in contrast, local redundancy bounds worst-case performance over all possible target distributions.

Several solutions to retain plasticity have also been proposed: shrink-and-perturb (Ash & Adams, 2020), continual backprop (Dohare et al., 2024), regenerative regularization (Kumar et al., 2023), and periodic resets (Nikishin et al., 2022). Separately, continual learning methods address catastrophic forgetting via replay (Rolnick et al., 2019), regularization such as EWC (Kirkpatrick et al., 2017), or architectural isolation (Rusu et al., 2016). Our contribution is orthogonal: rather than intervening on plasticity, we provide a principled measure grounded in universal compression theory that quantifies plasticity, predicting downstream performance better than existing metrics.

## 3. Local Redundancy as Plasticity

We formalize plasticity as the capacity of a network to learn new tasks from its current state. Our framework builds on universal compression theory, which provides a principled measure of model class complexity called *redundancy*.

### 3.1. Redundancy

Let $(X, Y) \sim P_{XY}$ with inputs $X \in \mathcal{X}$ and discrete labels $Y \in \mathcal{Y}$. Given $n$ samples $(x_i, y_i)$ drawn i.i.d. from $P_{XY}$, we model the conditional distribution using a parametric class $\{P_{Y|X,\theta} : \theta \in \Theta\}$. A *predictor* $Q_{Y^n|X^n}$ is any conditional distribution over label sequences.

The *worst-case redundancy* measures the excess log-loss of the best predictor relative to an oracle that knows the true parameter:

$$R_n^{**}(\Theta) \triangleq \min_Q \max_{x^n, y^n} \sup_{\theta \in \Theta} \log \frac{P_{Y^n|X^n,\theta}(y^n|x^n)}{Q(y^n|x^n)}. \quad (1)$$

This minimax redundancy equals $R_n^{**}(\Theta) = \sup_{x^n} \log Z(x^n)$, where the *Shtarkov sum*

$$Z(x^n) = \sum_{y^n \in \mathcal{Y}^n} \sup_{\theta \in \Theta} P_\theta(y^n|x^n) \quad (2)$$

sums the best-fit likelihood over all possible labelings (Shtarkov, 1987). Redundancy quantifies model class complexity: richer families incur higher redundancy because the predictor must hedge across more possible models.

### 3.2. Local Model Families

Global redundancy $R_n^{**}(\Theta)$ characterizes the entire model class and depends only on architecture. But plasticity is inherently *local*: it measures what a network can learn from its *current parameters* $\theta_0$—and hence its current input–output map—not what the architecture could learn in principle.

We stress that "local" here refers to this neighborhood in parameter space, not to local minima or local curvature of the loss. A network with high global capacity but collapsed representations has low plasticity.

To capture this, we define the *local model family* as the set of distributions reachable within an infinitesimal distance along gradient directions:

**Definition 3.1** (Local model family). Fix $\theta_0 \in \Theta$ and step size bound $\epsilon > 0$. Let $L(\theta; x^n, y^n) = -\log P_{Y^n|X^n,\theta}(y^n|x^n)$. The *local parameter set* is

$$\Theta(\theta_0, \epsilon) \triangleq \{\theta_0 - \eta\nabla_\theta L(\theta_0; x^n, y^n) : x^n \in \mathcal{X}^n, y^n \in \mathcal{Y}^n,$$
$$\eta \in [0, \epsilon]\}. \quad (3)$$

The *local model family* is $\mathcal{M}(\theta_0, \epsilon) = \{P_{Y^n|X^n,\theta} : \theta \in \Theta(\theta_0, \epsilon)\}$.

This captures all parameters reachable by one gradient step on *any* dataset $(x^n, y^n)$ with step size at most $\epsilon$. The gradient-based definition is more principled than a Euclidean ball $\{\theta : \|\theta - \theta_0\| < \epsilon\}$: perturbations in different directions have vastly different effects on predictions, and the gradient naturally weights directions by their influence on the loss (Zhang et al., 2019; Frankle et al., 2020).

The *local redundancy* is the worst-case redundancy of this family:

$$R_n^{**}(\theta_0, \epsilon) \triangleq R_n^{**}(\Theta(\theta_0, \epsilon)). \quad (4)$$

### 3.3. Plasticity as Local Redundancy

We argue that local redundancy is a natural measure of plasticity. While worst-case redundancy measures performance against the hardest distribution, average-case redundancy $R_n^*(\Theta) = \inf_Q \sup_\pi \mathbb{E}_{\theta \sim \pi}[\mathrm{KL}(P_\theta\|Q)]$ measures performance against a prior-weighted mixture. By the capacity-redundancy theorem (Haussler & Opper, 1997), average-case redundancy equals channel capacity:

$$R_n^*(\Theta) = \sup_{\pi \in \mathcal{P}(\Theta)} I(\theta; Y^n|X^n). \quad (5)$$

This quantity has a geometric interpretation: capacity equals the *information radius*, the radius of the smallest KL-divergence ball centered at some distribution $Q^*$ that contains all models $\{P_{Y|X,\theta} : \theta \in \Theta\}$ (Polyanskiy & Wu, 2025). A large information radius means models in $\Theta(\theta_0, \epsilon)$ are able to represent many diverse conditional distributions—a single gradient step can steer the network toward many targets. Low information radius means all locally reachable models have low KL divergence from each other. This matches plasticity in continual learning (Lyle et al., 2023; Dohare et al., 2024): a plastic network adapts readily, while a saturated network has collapsed to a region where gradients cannot induce meaningful change in the output distribution. Worst-case and average-case redundancy are asymptotically equal in leading term in classical and singular model

families (Watanabe, 2009), which motivates reading worst-case local redundancy through this geometric lens. We emphasize, however, that the information-radius interpretation in (5) serves as geometric *motivation*: our formal results (Theorems 3.2 and 3.4) concern worst-case local redundancy, and we do not claim a finite-sample worst-case/average-case equivalence for the infinitesimal local family.

### 3.4. Memorization Bounds for Classification

Next, we show that memorization provides an efficiently computable lower bound on local worst-case redundancy. Intuitively, we ask how much information about *random* targets a single infinitesimal update can encode: if many random labelings of fixed inputs can be fit after one small step, the local family contains many distinguishable conditional distributions and is therefore plastic (Figure 1). Sampling targets from the model's own predictive distribution $P_{\theta_0}$ is what turns this intuition into a lower bound, through the entropy cancellation in the proof of Theorem 3.4.

Fix inputs $x^n \in \mathcal{X}^n$ and let $\hat{\theta}_{y^n} \in \arg\max_{\theta \in \Theta} P_\theta(y^n|x^n)$ denote the MLE for labeling $y^n$. Define the *memorization gain*:

$$r(x^n; \Theta, \theta_0) \triangleq H(P_{\theta_0}(Y^n|x^n))$$
$$+ \mathbb{E}_{y^n \sim P_{\theta_0}(Y^n|x^n)}\left[\log P_{\hat{\theta}_{y^n}}(y^n|x^n)\right]. \quad (6)$$

**Theorem 3.2** (Memorization lower bound)**.** *For any inputs* $x^n$ *and model class* $\Theta$,

$$R_n^{**}(\Theta) \geq r(x^n; \Theta, \theta_0). \quad (7)$$

*Proof.* Recall $R_n^{**}(\Theta) = \sup_{x^n} \log Z(x^n)$ where $Z(x^n) = \sum_{y^n} \sup_\theta P_\theta(y^n|x^n)$. For any fixed $x^n$ and any distribution $Q$ over $\mathcal{Y}^n$:

$$\log Z(x^n) = \log \sum_{y^n \in \mathcal{Y}^n} P_{\hat{\theta}_{y^n}}(y^n|x^n)$$
$$= \log \sum_{y^n} Q(y^n)\frac{P_{\hat{\theta}_{y^n}}(y^n|x^n)}{Q(y^n)}$$
$$\geq \mathbb{E}_{y^n \sim Q}\left[\log \frac{P_{\hat{\theta}_{y^n}}(y^n|x^n)}{Q(y^n)}\right] \quad (8)$$
$$= H(Q) + \mathbb{E}_{y^n \sim Q}[\log P_{\hat{\theta}_{y^n}}(y^n|x^n)],$$

where (8) applies Jensen's inequality to the concave logarithm. Taking $Q = P_{\theta_0}(\cdot|x^n)$ yields $\log Z(x^n) \geq r(x^n; \Theta, \theta_0)$. □

For local families with small $\epsilon$, the MLE is achieved by moving distance $\epsilon$ along the gradient direction of the given labeling.

**Proposition 3.3** (Local MLE)**.** *Fix inputs* $x^n$ *and labels* $y^n$. *Let* $\hat{\theta}_{y^n}(\epsilon) \in \arg\max_{\theta \in \Theta(\theta_0,\epsilon)} P_\theta(y^n|x^n)$. *Then as* $\epsilon \to 0$:

$$\hat{\theta}_{y^n}(\epsilon) = \theta_0 - \epsilon \nabla_\theta L(\theta_0; x^n, y^n) + O(\epsilon^2), \quad (9)$$
$$L(\hat{\theta}_{y^n}; x^n, y^n) = L(\theta_0; x^n, y^n) - \epsilon\|\nabla_\theta L(\theta_0; x^n, y^n)\|^2$$
$$+ O(\epsilon^2).$$

This follows from Taylor expansion: the gradient step maximally decreases loss to first order. Combining with Theorem 3.2 yields a characterization of local redundancy in terms of gradient norms.

**Theorem 3.4** (Local redundancy via gradients)**.** *For the local family* $\Theta(\theta_0, \epsilon)$ *with sufficiently small* $\epsilon$,

$$R_n^{**}(\theta_0, \epsilon) \geq \epsilon \cdot \mathbb{E}_{y^n \sim P_{\theta_0}}\left[\|\nabla_\theta L(\theta_0; x^n, y^n)\|^2\right]$$
$$+ O(\epsilon^2). \quad (10)$$

*Proof.* By Proposition 3.3, averaging over $y^n \sim P_{\theta_0}$:

$$\mathbb{E}_{y^n}[\log P_{\hat{\theta}_{y^n}}(y^n|x^n)]$$
$$= \mathbb{E}_{y^n}[\log P_{\theta_0}(y^n|x^n)] + \epsilon \mathbb{E}_{y^n}\left[\|\nabla_\theta L\|^2\right] + O(\epsilon^2)$$
$$= -H(P_{\theta_0}) + \epsilon \mathbb{E}_{y^n}\left[\|\nabla_\theta L\|^2\right] + O(\epsilon^2). \quad (11)$$

Substituting into (6), the entropy terms cancel, giving $r = \epsilon \mathbb{E}_{y^n}[\|\nabla_\theta L\|^2] + O(\epsilon^2)$. The result follows from Theorem 3.2. □

### 3.5. Memorization Bounds for Regression

The preceding results assume discrete labels $\mathcal{Y}$. We now show that our methods generalize to regression settings, where labels are continuous, using rate-distortion theory. This extension justifies measuring local redundancy via gradient norms on random Gaussian targets—precisely the setup used in our time series experiments.

**Proposition 3.5** (Regression memorization bound)**.** *Consider regression with MSE loss. Let* $Q = \mathcal{N}(\mu, \sigma_y^2 I)$ *be any target distribution over* $\mathbb{R}^m$ *with arbitrary mean* $\mu \in \mathbb{R}^m$. *If model family* $\Theta$ *achieves expected MSE* $= D$ *when fitting* $n$ *i.i.d. samples* $y_i \sim Q$, *then*

$$R_n^{**}(\Theta) \geq \frac{nm}{2}\log\frac{\sigma_y^2}{D}. \quad (12)$$

This follows from the Gaussian rate-distortion theorem: representing $\mathcal{N}(\mu, \sigma_y^2)$ samples to MSE distortion $D$ requires rate $R(D) = \frac{1}{2}\log(\sigma_y^2/D)$ bits per scalar (Cover & Thomas, 2006). Discretizing targets and applying Theorem 3.2 to the induced classification problem yields the bound in the limit of fine quantization.

**(a) Vision: Synthetic images with random labels**

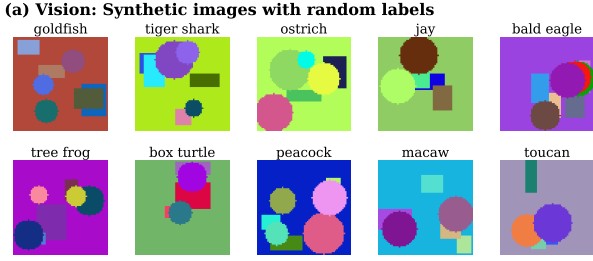

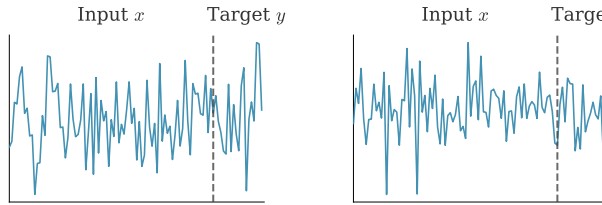

**(b) Time series: i.i.d. Gaussian inputs and targets**

*Figure 2.* Synthetic memorization datasets. **(a)** Vision classification uses input images consisting of randomly generated overlapping shapes on random backgrounds, paired with labels sampled from the model's predictive distribution. **(b)** Time series regression uses i.i.d. Gaussian inputs and targets with no temporal structure, sampled from the model's predictive distribution. By Theorem 3.2, any choice of inputs $x^n$ and properly randomized targets yields an estimator that lower bounds the worst-case redundancy. We design inputs to make memorization easier for model parameters to express, yielding a smoother and stronger lower bound on local redundancy.

**Corollary 3.6** (Gradient bound for regression). *Theorem 3.4 extends to MSE loss. Let $P_{\theta_0}(y|x) = \mathcal{N}(f_{\theta_0}(x), \sigma^2 I)$. Then:*

$$R_n^{**}(\theta_0, \epsilon) \geq \epsilon \cdot \mathbb{E}_{y^n \sim P_{\theta_0}} \left[ \|\nabla_\theta L(\theta_0; x^n, y^n)\|^2 \right] + O(\epsilon^2). \tag{13}$$

The gradient norm measures the rate (in bits per unit step size) at which the model can encode information about random targets.

### 3.6. Synthetic Memorization Datasets

Theorems 3.2, 3.4, and Corollary 3.6 reduce estimating local redundancy to measuring gradient norms on synthetic data with random labels (or regression targets). We emphasize that the model is never trained on this synthetic data: the synthetic batch is used only to probe the local family and evaluate the gradient-norm estimator, leaving the parameters $\theta_0$ unchanged. The randomized-target rule $y \sim P_{\theta_0}$ is also essential—it is what makes the gradient norm a lower bound on local redundancy rather than a generic gradient-sensitivity heuristic. The quality of this lower bound depends only on how well the synthetic data saturates model capacity; we ablate the choice of synthetic inputs in Ap-

*Table 1.* Correlation ($\times 100$) between each plasticity metric and average future task accuracy ($t + 1, \ldots, t + 10$), after regressing out task number. Values are correlations with the residual accuracy. Local redundancy attains the strongest residual correlation for both Pearson and Spearman.

| Plasticity Metric | Pearson $r$ (res.) | Spearman $\rho$ (res.) |
|---|---|---|
| **Local redundancy** | $\mathbf{8.9 \pm 0.4}$ | $\mathbf{8.8 \pm 0.4}$ |
| Distance from init | $7.3 \pm 0.4$ | $5.9 \pm 0.4$ |
| Weight norm | $4.6 \pm 0.4$ | $7.1 \pm 0.4$ |
| Dormant neuron ratio | $3.4 \pm 0.5$ | $4.0 \pm 0.4$ |
| Training grad. norm | $1.6 \pm 0.4$ | $5.0 \pm 0.4$ |

pendix A.4. Since local redundancy is intractable to compute directly, we use this lower bound as a proxy for it throughout our experiments.

**Vision classification.** We generate synthetic images by: (i) filling a canvas with a uniform random RGB background; (ii) overlaying randomly positioned rectangles with random colors; and (iii) overlaying randomly positioned circles with random colors. For each image $x$, we sample the label from the model's predictive distribution $y \sim P_{\theta_0}(y|x)$. These images are visually diverse but contain no learnable structure, lower bounding local redundancy by Theorem 3.2.

**Time series regression.** We generate random Gaussian context windows $x \sim \mathcal{N}(0, I)$ of shape (context length $\times$ features). Next, prediction targets are sampled from the model's predictive distribution $y \sim \mathcal{N}(f_{\theta_0}(x), \sigma^2 I)$. By Corollary 3.6, the squared gradient norm on this data lower bounds local redundancy.

## 4. Experiments

We validate our theoretical framework in two settings: continual image classification and time series transfer learning. In continual learning, we show that local redundancy decreases during training and predicts future task performance better than existing metrics. In time series transfer, we demonstrate that local redundancy identifies optimal pretraining checkpoints when validation loss plateaus. Throughout, we estimate local redundancy via Theorem 3.4: the mean squared gradient norm on synthetic data with random labels sampled from the model's predictive distribution.

### 4.1. Continual Learning

**Setup.** We evaluate on Continual ImageNet (Dohare et al., 2024), a continual learning benchmark derived from ImageNet (Deng et al., 2009). The benchmark constructs a sequence of binary classification tasks by pairing randomly selected ImageNet classes. With 1000 ImageNet classes, there are approximately 500,000 possible pairs; we uniformly sample 3000 tasks without replacement. Because

each task involves different classes drawn from the same distribution, task difficulty remains constant in expectation, so any systematic decline in performance reflects loss of plasticity.

**Protocol.** We train MobileNetV3-Large (Howard et al., 2019), a convolutional network, on these tasks sequentially without access to previous task data: after training and evaluation on one task, the network proceeds to the next with weights carried over but no replay of earlier examples. We estimate local redundancy at each task boundary via Theorem 3.4. We generate $n = 5000$ synthetic images as described in Section 3.6, each consisting of 5 rectangles and 5 circles drawn in random order with random sizes, positions, and RGB colors on a random RGB background, with labels sampled from the model's predictive distribution. We compute the mean squared gradient norm of the cross-entropy loss with respect to all parameters, averaged over the synthetic batch. Full training details are provided in Appendix A.

**Baselines.** We compare against four existing plasticity metrics: weight norm, distance from initialization, dormant neuron ratio (Sokar et al., 2023), and training gradient norm. Weight norm, distance from initialization, and dormant neuron ratio capture existing measures of plasticity. Training gradient norm computes gradient magnitude on real task data rather than synthetic memorization data; comparing against this baseline isolates the contribution of our synthetic data design.

**Results.** Figure 3 shows local redundancy and test accuracy across the task sequence. Local redundancy declines steadily as training progresses, consistent with progressive plasticity loss as the network commits to learned mappings. To test whether this decline predicts future performance, we correlate each metric at task $t$ with average accuracy over tasks $t + 1, \ldots, t + 10$. Because accuracy drifts downward due to catastrophic forgetting, task number alone is highly predictive: a metric that simply decreases over time would correlate with future accuracy without capturing useful signal. To isolate genuine predictive signal above overall downward drift of performance, we first regress out task number and measure correlation with the residuals. Future task accuracy is inherently noisy: some class pairs are easy to distinguish (e.g., animals vs. vehicles) while others are difficult (e.g., two similar dog breeds), and tasks are sampled randomly. Still, Table 1 shows that local redundancy attains a meaningful residual correlation—the highest across all plasticity metrics for both Pearson and Spearman.

**Stability–plasticity tradeoff.** We also ask how local redundancy relates to forgetting. For each task we measure the drop in a previous task's accuracy after training on the next (positive values indicate more forgetting), again regressing out task number. Table 2 reports the correlation of each

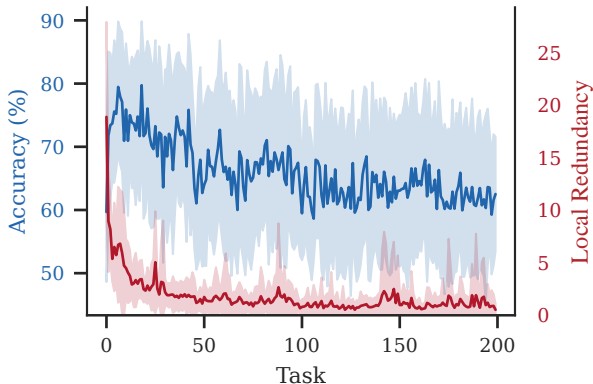

*Figure 3.* Local redundancy (red) and test accuracy (blue) over 200 sequential binary classification tasks on Continual ImageNet. Each task consists of distinguishing a randomly selected pair of ImageNet classes, with training proceeding sequentially without re-initialization. Local redundancy decreases as the network commits to learned mappings, tracking the expected loss of plasticity.

*Table 2.* Correlation ($\times 100$) between each plasticity metric and forgetting (the drop in a previous task's accuracy after training on the next task; positive indicates more forgetting), after regressing out task number. Local redundancy is positively associated with forgetting and among the largest such correlations, consistent with the stability–plasticity tradeoff.

| Plasticity Metric | Pearson $r$ (res.) | Spearman $\rho$ (res.) |
|---|---|---|
| **Local redundancy** | $13.0 \pm 4.5$ | $23.5 \pm 4.4$ |
| Distance from init | $-2.0 \pm 4.5$ | $3.7 \pm 4.5$ |
| Weight norm | $2.2 \pm 4.5$ | $4.1 \pm 4.5$ |
| Dormant neuron ratio | $-5.3 \pm 4.5$ | $-2.9 \pm 4.5$ |
| Training grad. norm | $6.2 \pm 4.5$ | $23.6 \pm 4.4$ |

metric with this forgetting signal. Local redundancy is positively correlated with forgetting and among the largest such correlations across all metrics. This is consistent with interpreting it as plasticity: a network that retains more capacity to adapt also overwrites prior tasks more readily, recovering the stability–plasticity tradeoff.

### 4.2. Time Series Transfer Learning

We apply local redundancy to checkpoint selection for time series forecasting, where transfer learning is common due to limited labeled data in target domains. With limited data, selecting pretrained checkpoints is difficult—models often overfit to the source domain and fail to generalize to the target. We show that local redundancy can be used to select pretrain checkpoints with higher downstream performance by identifying checkpoints that have higher plasticity.

**Setup.** We evaluate on the ETT (Electricity Transformer Temperature) benchmark (Zhou et al., 2021), a standard testbed for time series transfer learning. The dataset contains

*Table 3.* Average fine-tuning validation MSE on ETTh2 across checkpoint selection strategies (120 seeds, mean ± stderr). Selecting the checkpoint with maximum local redundancy achieves better downstream performance than lowest validation loss: local redundancy identifies checkpoints that retain capacity to adapt, even when validation loss has plateaued. Local redundancy is the only plasticity metric with this property, outperforming dormant neuron ratio and weight norm. Linear probing outperforms local redundancy but requires significantly more compute and access downstream labels, making it inadmissible when the target domain is unknown at selection time.

| Selection Strategy | Finetune Val Loss |
|---|---|
| Linear probing[†] | $0.1918 \pm 0.0005$ |
| **Max local redundancy** | **$0.1937 \pm 0.0005$** |
| Lowest val. loss | $0.1952 \pm 0.0006$ |
| Min dormant ratio | $0.1960 \pm 0.0005$ |
| Min weight norm | $0.1968 \pm 0.0004$ |
| No pretraining | $0.2168 \pm 0.0079$ |

[†]Checkpoint selection requires access to downstream task.

measurements from electricity transformers stations, including oil temperature and six power load features recorded over two years. We pretrain on ETTm1 (15-minute readings from station 1) and finetune on ETTh2 (hourly readings from station 2). This setup represents realistic domain shift: the datasets share the same features but differ in temporal granularity and station-specific dynamics. The task is multivariate forecasting: given a context window, predict future values across all features.

**Checkpoint selection strategies.** The standard approach selects the checkpoint with lowest pretraining validation loss. We compare against selecting by maximum local redundancy, as well as two other proxies: minimum dormant neuron ratio (Sokar et al., 2023) and minimum weight norm. As a strong alternative, we use linear probing (Alain & Bengio, 2017): for each checkpoint, we freeze all layers except the final prediction head, randomly reinitialize the head, and train it for one epoch on labeled data from the target domain; we then select the checkpoint with lowest probe loss. Linear probing requires downstream labels and a small training run conducted at each checkpoint. We also compare against training from scratch (no pretraining).

**Protocol.** We pretrain for 12 epochs with learning rate $2 \times 10^{-3}$ with cosine annealing and perform checkpoint selection according to each strategy described above. We estimate local redundancy by computing the mean squared gradient norm on synthetic data generated using the protocol in Section 3.6. We compute this metric four times per epoch during pretraining and average across measurements. For each chosen checkpoint, we fine-tune for 2 epochs or until finetuning validation loss saturates, with learning rate $2 \times 10^{-4}$. We repeat the entire pipeline across 120 random seeds and report the mean and standard error of the final

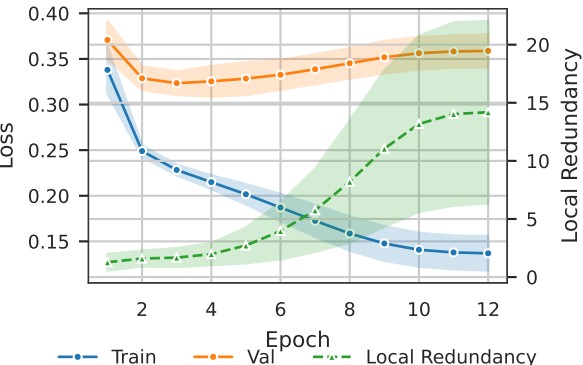

*Figure 4.* Training loss (blue), validation loss (orange), and local redundancy (green) during pretraining on ETTm1 (shading indicates standard deviation across seeds). Validation loss saturates after 3-6 epochs, yet local redundancy continues to increase up to 12 epochs, peaking at different epochs for different seeds. The network's capacity to adapt continues to evolve after validation performance saturates: selecting checkpoints with large local redundancy takes advantage of this and leads to stronger downstream performance.

fine-tuning validation loss.

**Results.** Table 3 shows fine-tuning performance for each checkpoint selection strategy, averaged across 120 total seeds. Selecting the checkpoint with maximum local redundancy achieves better downstream performance than selecting by lowest validation loss: once validation loss plateaus, it no longer meaningfully discriminates between checkpoints—they all fit the pretraining distribution adequately. However, local redundancy continues to vary, revealing differences in remaining adaptability. Figure 5 shows that checkpoints with higher local redundancy adapt faster during fine-tuning, despite starting from a farther initial distribution. Additionally, local redundancy is the only plasticity metric with this property, outperforming dormant neuron ratio and weight norm. Linear probing outperforms local redundancy but requires downstream labels and a small training run conducted at each checkpoint, making it inefficient and inadmissible when the target domain is unknown at selection time.

### 4.3. Memorization Capacity

Theorem 3.4 provides a lower bound on local redundancy. For this bound to yield a useful plasticity estimate, the gradient norm must behave smoothly over training. To verify this on our designed synthetic datasets, we train to convergence on synthetic data and inspect the bits per parameter memorized.

**Setup.** We generate an image classification and a time series regression synthetic dataset as described in Section 3.6. For classification, bits memorized follows from cross-entropy

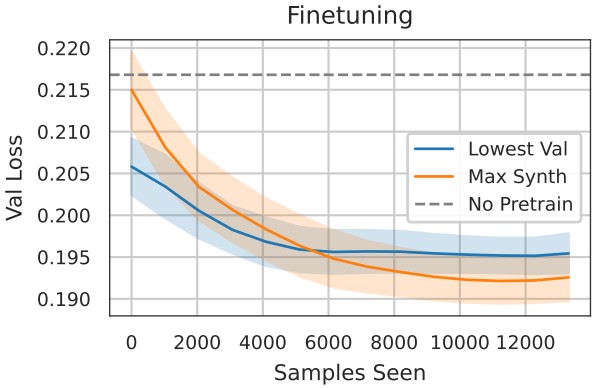

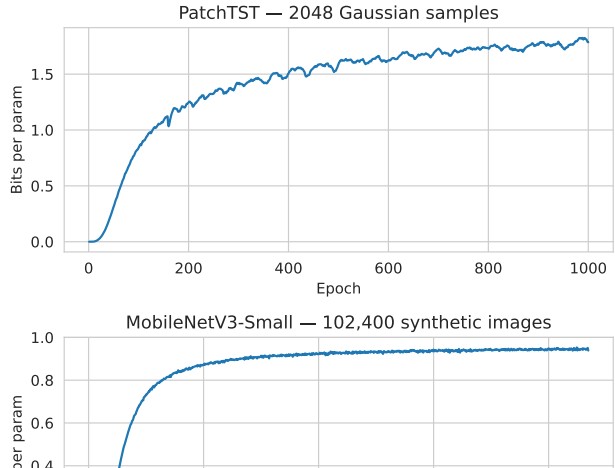

*Figure 5.* Fine-tuning validation loss on ETTh2 for checkpoints selected by lowest pretraining validation loss (blue) versus maximum local redundancy (orange). The high-redundancy checkpoint begins with higher loss but adapts more quickly, achieving a lower final loss on average (shading indicates standard deviation across seeds). Dashed line shows the final validation loss when training from scratch.

*Figure 6.* Bits per parameter over training on synthetic memorization data. The time series model (top) achieves 1.7 bits per parameter and the vision model (bottom) achieves 1.0. The stability in the early epochs of training confirms that our gradient-based estimator provides a stable estimate for local redundancy.

loss as given by Theorem 3.2; for regression, from rate-distortion theory (Corollary 3.6). We train in bf16 precision and report bits per parameter: total bits memorized divided by the number of parameters.

**Results.** Figure 6 shows bits per parameter over training for each experiment. The time series model saturates at 1.7 bits per parameter and the vision model saturates at 1.0 bits per parameter. Prior works on memorization capacity report similar rates, such as 2.0 bits per parameter for large GPT-style transformers Allen-Zhu & Li (2024). The ability of neural networks to utilize free parameters to memorize patterns is precisely why synthetic memorization can provide non-vacuous lower bounds on local redundancy. Additionally, the increase in bits per parameter increases smoothly during the early stages of training, indicating that the gradient norm in Section 3.6 is indeed a stable relative estimate of local redundancy when applied to a model that has not seen the synthetic data during training.

## 5. Conclusion

We introduced local redundancy, an information-theoretic measure of plasticity grounded in universal compression theory. We define local redundancy as the worst-case redundancy of a local model family—the set of parameters reachable by a single gradient step—to obtain a principled quantity with the correct geometric meaning: the information radius of locally achievable distributions. Theorem 3.4 lower bounds local redundancy with the gradient norms on any inputs with properly randomized targets, requiring only a single backward pass and no access to downstream tasks, and Section 4.3 verifies that the synthetic gradient norm

provides a stable estimate of local redundancy.

Our experiments validate the framework in two settings. In continual learning, local redundancy tracks plasticity loss and predicts future task accuracy better than existing metrics. In transfer learning, local redundancy identifies high-quality checkpoints when validation loss has plateaued.

**Limitations.** Our results establish that local redundancy *predicts* downstream performance, but the lower bound does not establish that it *causes* it; a controlled study that intervenes on plasticity directly—for example via replay, EWC, or shrink-and-perturb—is needed to test causation. Local redundancy is also a deliberately one-step, local quantity: it characterizes the family reachable by a single gradient step and is not a global account of plasticity loss over long optimization horizons. Our experiments measure forward adaptability rather than the full stability–plasticity tradeoff, and are limited to models of up to a few million parameters. Finally, because local redundancy is intractable, we use the gradient-norm lower bound as a proxy for it; this proxy is not calibrated to the true value and could mislead where the bound is loose, for instance if the gap between the bound and true redundancy varies across the checkpoints being compared. Consistent with this, our synthetic data saturates only approximately 1–2 bits per parameter (Section 4.3), short of the theoretical ceiling of 16 bits per parameter.

**Future work.** Promising directions include validating scaling behavior on larger models, constructing better synthetic memorization datasets to saturate model capacity further, and developing interventions that preserve local redundancy during training.

## Impact Statement

This paper presents work whose goal is to advance the field of machine learning. There are many potential societal consequences of our work, none of which we feel must be specifically highlighted here.

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

# A. Implementation Details

## A.1. Continual Learning

**Architecture.**    We use MobileNetV3-Large (Howard et al., 2019), a convolutional network with 5.47M parameters. We replace the final classification head with a two-class output layer for binary classification. The model is trained in bfloat16 precision.

**Dataset.**    We evaluate on Continual ImageNet (Dohare et al., 2024), constructing 3000 binary classification tasks by sampling pairs of ImageNet classes uniformly without replacement. Images are resized to 224×224 and normalized using ImageNet statistics.

**Training.**    For each task, we train for 100 epochs using AdamW with learning rate $10^{-3}$. Training is conducted on 4 H200 GPUs with an effective batch size of 1024 ($256 \times 4$). After training on each task, we evaluate on held-out examples from the same class pair, then proceed to the next task with weights carried over. No replay or regularization is used.

**Local redundancy estimation.**    At each task boundary, we estimate local redundancy via Theorem 3.4. We generate $n = 5000$ synthetic images, each consisting of 5 rectangles and 5 circles drawn in random order with random sizes, positions, and RGB colors on a random RGB background. For each image $x$, we sample the label from the model's predictive distribution $y \sim P_{\theta_0}(y|x)$. We compute the mean squared gradient norm of the cross-entropy loss with respect to all parameters, averaged over the synthetic batch.

**Baseline metrics.**    We compute the following plasticity metrics at each task boundary:

   (i) *Weight norm*: total $L_2$ norm of all parameters.
  (ii) *Distance from initialization*: $L_2$ distance $\|\theta - \theta_{\text{init}}\|$.
 (iii) *Dormant neuron ratio*: fraction of ReLU neurons with zero activation on a batch of real training data.
 (iv) *Training gradient norm*: mean squared gradient norm on real task data (rather than synthetic).

## A.2. Time Series Transfer Learning

**Architecture.**    We use PatchTST (Nie et al., 2023), a Transformer-based model for multivariate time series forecasting. The model segments each input channel into patches of length 16 with stride 8, then processes them through 4 Transformer encoder layers with $d_{\text{model}} = 256$, 8 attention heads, FFN dimension 512, and dropout 0.1. The context length is 512 timesteps and the prediction horizon is 96 timesteps across all 7 input channels. The model has approximately 2.26M parameters and is trained in bfloat16 precision.

**Datasets.**    We pretrain on ETTm1 and fine-tune on ETTh2 (Zhou et al., 2021). Both datasets contain 7 features recorded from electrical transformers. Each dataset is split 80/20 into train and validation sets. We construct sliding window samples of length $512 + 96$ and normalize each feature to zero mean and unit variance using training set statistics.

**Pretraining.**    We train with AdamW for 12 epochs using learning rate $2 \times 10^{-3}$ with cosine annealing. Training uses DataParallel across 4 H200 GPUs with an effective batch size of 1024 ($256 \times 4$). We save checkpoints after every epoch and compute local redundancy at four evenly spaced points within each epoch. Local redundancy is estimated as the mean squared gradient norm on a synthetic dataset of 1024 random Gaussian inputs $x \sim \mathcal{N}(0, I)$ with targets sampled from the model's predictive distribution $y \sim \mathcal{N}(f_{\theta_0}(x), \sigma^2 I)$.

**Checkpoint selection.**    We evaluate several strategies for selecting which pretrained checkpoint to fine-tune:

   (i) *Lowest validation loss*: select the checkpoint with lowest pretraining validation MSE.
  (ii) *Maximum local redundancy*: select the checkpoint with highest local redundancy.
 (iii) *Linear probe*: freeze the encoder and train only the prediction head for one epoch on target domain data; select the checkpoint with lowest probing loss.
 (iv) *Minimum weight norm*: select the checkpoint with smallest total $L_2$ parameter norm.
  (v) *Minimum dormant ratio*: select the checkpoint with fewest dormant FFN neurons, where a neuron is dormant if its mean activation on pretraining data falls below 0.05.

**Fine-tuning.** From each selected checkpoint, we fine-tune on ETTh2 for 2 epochs with learning rate $2 \times 10^{-4}$, using the same DataParallel configuration. We perform 6 fine-tuning runs per checkpoint selection strategy per pretrain run and report the mean final validation MSE. Results are aggregated across 120 pretrain runs.

### A.3. Memorization Capacity

**Time series experiment.** We train PatchTST (2.26M parameters, bfloat16) to memorize 2048 samples of i.i.d. Gaussian noise. Each sample consists of 512 input timesteps and 96 target timesteps across 7 channels, yielding 1,376,256 total output values to memorize. We train for 1000 epochs using AdamW with learning rate $10^{-3}$. We conduct training on 4 H200 GPUs with an effective batch size of 1024 ($256 \times 4$).

**Vision experiment.** We train MobileNetV3-Large (5.47M parameters, bfloat16) to memorize 819,200 synthetic images with random class labels from 1000 classes. Each image consists of 5 rectangles and 5 circles with random colors, sizes, and positions drawn in random order on a random RGB background. We train for 1000 epochs using AdamW with learning rate $10^{-3}$. We conduct training on 4 H200 GPUs with an effective batch size of 1024 ($256 \times 4$).

### A.4. Synthetic Data Ablation

By Theorem 3.2, any inputs with properly randomized targets yield a valid lower bound on local redundancy; the choice of synthetic inputs affects only the *tightness* of that bound. To assess sensitivity to this choice, we compare three input types for the vision estimator: (i) independent per-pixel Gaussian noise, (ii) our synthetic shapes (Section 3.6), and (iii) real, unseen images from the next task. The real-image variant serves as a reference but is inadmissible in general, since it requires access to the future task; we therefore measure how closely the Gaussian-noise and shape-based estimates track it across training. Table 4 reports the agreement between each synthetic estimate and the real-image reference, and Figure 7 plots the three estimates over the task sequence. The shape-based inputs track the reference closely, whereas per-pixel Gaussian noise produces smaller gradients and a noisier, erratic estimate. All variants remain valid lower bounds; the shape construction simply yields a stronger and smoother one. For the time series estimator, we found that Gaussian inputs already track the reference well and adopt them throughout.

*Table 4.* Agreement between the vision local-redundancy estimate computed on synthetic inputs and the estimate computed on real, unseen next-task images (reference), measured across training. The shape-based construction tracks the reference far more closely than per-pixel Gaussian noise.

| Synthetic input | Pearson $r$ | $R^2$ | MSE |
|---|---|---|---|
| Shapes | 0.964 | 0.929 | 0.067 |
| Gaussian noise | 0.847 | 0.718 | 2.474 |

### A.5. Computational Cost

Estimating local redundancy requires a single backward pass on the synthetic batch, with no auxiliary optimization or downstream supervision. Table 5 reports wall-clock cost on a 500-task Continual ImageNet run (one H100, one measurement per task): it adds only $1.65\%$ of total training time, more than the simplest metrics but far below the dormant neuron ratio.

*Table 5.* Wall-clock cost of computing each plasticity metric on a 500-task Continual ImageNet run (one H100, one measurement per task). Training takes 81 minutes; local redundancy adds $1.65\%$ of total training time.

| Plasticity Metric | Total (s) | Per-task (s) |
|---|---|---|
| **Local redundancy** | 80.3 | 0.161 |
| Distance from init | 3.0 | 0.006 |
| Weight norm | 1.5 | 0.003 |
| Training grad. norm | 1.2 | 0.003 |
| Dormant neuron ratio | 290.1 | 0.580 |

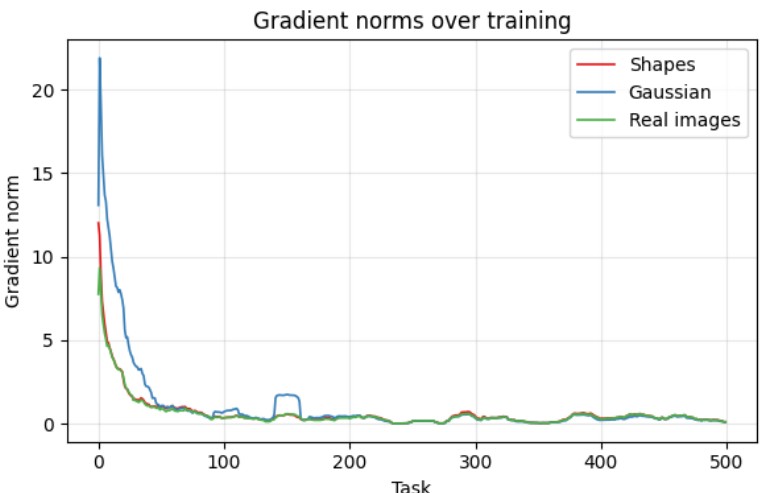

*Figure 7.* Local-redundancy estimate (mean squared gradient norm) over the Continual ImageNet task sequence, computed on three input types: synthetic shapes, per-pixel Gaussian noise, and real, unseen next-task images (reference). The shape-based estimate tracks the real-image reference closely, whereas per-pixel Gaussian noise is noisier and erratic (e.g., the spike near task 150).

