# OpenReview forum: "Local Redundancy: An Information-Theoretic Measure of Plasticity from Synthetic Memorization"
_ICML.cc/2026/Conference — ICML 2026 spotlight_

### Official Review · Reviewer_5Fm6 · 2026-02-20

**Soundness:** 2
**Presentation:** 3
**Significance:** 3
**Originality:** 3
**Overall Recommendation:** 4
**Confidence:** 3

**Summary:**

In this paper, the authors introduce a scalar measure of plasticity based on local minimax redundancy. Specifically, the measure captures the maximally diverse local behaviors possible across inputs caused by taking an infinitesimally small step in parameter space, via a one-backward-pass lower bound. Intuitively, this quantifies how much the model's input–output mapping can change under a small parameter perturbation in the worst case over datasets — thereby serving as an indicator of plasticity.

The authors demonstrate that their measure is the most consistently correlated with future task accuracy on the Continual ImageNet dataset. They further show that the measure can be used to inform transfer learning on time-series data and to guide checkpoint selection.

**Compliance With Llm Reviewing Policy:**

Affirmed.

**Final Justification:**

Happy to accept the paper, I was unaware of the time required for experiments in the continual learning case. Also, the authors addressed all my remaining concerns in the rebuttals.

**Key Questions For Authors:**

See weaknesses.

**Limitations:**

yes

**Strengths And Weaknesses:**

## Strengths

The core idea is elegant and well-motivated through the existing literature. The ability to distill something as complex as plasticity into a single scalar quantity — and to show that it correlates meaningfully with downstream performance — is a strong result. The theoretical grounding of the paper is particularly compelling.

## Weaknesses

The main weakness is that the evaluation is not comprehensive enough, and this concern falls along two major axes.

### Limited Model and Domain Coverage

For both the ImageNet and time-series case studies, only a single small model is investigated MobileNetV3-Large (5.47M parameters) and PatchTST (2.26M parameters), respectively. This leaves me unconvinced that the method would continue to perform well across models of varying size and architecture. Demonstrating the measure's effectiveness on a broader range of models would substantially strengthen the claims.

### Insufficient Empirical Justification

Several natural questions are left unanswered, and the paper would benefit from additional experiments to address them:

- **Computational cost:** The proposed method appears more involved than the baselines, which are generally quite simple. A comparison of computational overhead would be informative.
- **Ablation of the synthetic data:** The structure in the random data (circles and squares) is not clearly justified. How sensitive are the results to the choice of synthetic images? An ablation varying these inputs would help clarify their role.
- **Isolating the effect of the synthetic data:** Much of the predictive power may simply arise from introducing synthetic data and checking how easily the model can fit to it. Could an even simpler synthetic-gradient sensitivity metric (without the minimax redundancy framing and without the specific label-sampling scheme) match the predictive power? Attempting to disentangle the contribution of the synthetic data from the contribution of the minimax redundancy formulation would meaningfully strengthen the paper.

---

> ### Author Rebuttal · Authors · 2026-03-30
>
> Thank you for your careful review! We agree that broader empirical coverage and more ablations would strengthen the paper. At the same time, we think the main issue here is scope rather than correctness. The paper’s intended claims are narrower: (i) local redundancy is a formally defined local measure of plasticity, (ii) Theorem 3.4 gives a tractable first-order lower bound via synthetic memorization gradients, and (iii) the resulting estimator is useful in two distinct settings. Those are the claims the current paper is meant to support.
>
> Q1: Why only one model per domain, and can the method generalize more broadly?
> We agree that more architectures and scales would broaden the empirical story. Our intent here was not to make a universality claim across all model families and scales, but to test whether the same quantity remains useful across substantially different settings. The two case studies differ along several axes: CNN versus Transformer, classification versus regression, and continual learning versus transfer learning. Also, plasticity experiments are unusually expensive relative to ordinary training, because one often needs to train well past ordinary saturation, across long horizons and many seeds, before plasticity effects become stable. Our model scale is therefore modest, but comparable to prior empirical work on loss of plasticity.
>
> Q2: What is the computational cost?
> The estimator is just one ordinary backward pass on a synthetic batch at each measurement point. It does not introduce an auxiliary optimization problem, a second training loop, or downstream supervision. We initially did not include a separate wall-clock table because the extra cost is simply the standard cost of computing gradients during training, but we see that it may be confusing seeing multiple methods. The more important compute contrast in the paper is linear probing, which performs better but requires downstream labels plus an additional training run at every candidate checkpoint. We will add the compute times for each method in a revision.
>
> Q3: Why use circles and squares, and how sensitive is the method to the synthetic data?
> By Theorem 3.2, any chosen inputs $x^n$ together with properly randomized targets yield a valid lower bound. The role of the synthetic data is only to affect the strength of that lower bound. We started by defaulting to simple Gaussian noise, but for images, we found that simple pixelwise Gaussian noise gave smaller gradients and therefore a weaker estimate, while simple shape-based images produced a stronger and smoother bound. For time series, a Gaussian construction worked well. So the synthetic design is only to increase the quality of the lower bound. We agree the paper should explain this more clearly, and will revise the writing to do so.
>
> Q4: Is the predictive signal just coming from synthetic data rather than the minimax redundancy formulation?
> The estimator is not just a generic synthetic-gradient sensitivity score. The randomized target rule $y\sim P_{\theta_0}(y\mid x)$ is essential because it is what yields the entropy cancellation in Eq. (11), and therefore what makes the gradient norm a first-order lower bound to local redundancy via Theorems 3.2 and 3.4. Without that construction, one gets a different heuristic, not the same theoretically motivated quantity. Also, we would like to clarify that the model is not trained on the synthetic data (and will revise to make sure this is clear). The synthetic batch is used only to probe the local quantity and estimate the bound. It is not used to update the model in the usual synthetic-data sense. Any choice of inputs yields a valid lower bound, and choosing meaningful synthetic data happens to yield a stronger lower bound.
>
> Thank you again for the detailed feedback! We agree the empirical story could be broader, but we think these are breadth questions rather than evidence of a flaw in the formulation or the theoretical connection established by the paper.

---

> > ### Author Rebuttal · Reviewer_5Fm6 · 2026-04-02
> >
> > see comment.

---

> > > ### Author Response · Authors · 2026-04-05
> > >
> > > Thank you for the updated comments and feedback! We address each remaining point below.
> > >
> > > **Limited model and domain coverage.** We agree that the model sizes we investigate are relatively small. However, continual learning experiments are unusually expensive, sometimes requiring training orders of magnitude more than what is typical for a model (e.g., training to saturation for binary image classification tasks 500 times in Continual ImageNet) and train for many seeds to measure statistical significance. Models of this size are the most we can do with our compute budget of a few hundred H100 GPU hours. We believe they are comparable to those used in related work (e.g., Dohare et al., 2024). Additionally, we chose two diverse domains (images and time series) and architectures (CNN and Transformer) to try to cover as much ground as possible, despite these compute constraints. Finally, we note that local redundancy is theoretically motivated and grounded, rather than empirically driven, and we believe the consistency of results across these diverse settings, even at a smaller scale, provides meaningful validation.
> > >
> > > **Computational cost.** We measured wall-clock times for a single 500-task Continual ImageNet run on one H100 GPU, with one metric computation per task (training itself takes 81 minutes):
> > >
> > > | Metric | Total (s) | Per-task (s) |
> > > |---|---|---|
> > > | Local redundancy | 80.3 | 0.161 |
> > > | Weight norm | 1.5 | 0.003 |
> > > | Distance from init | 3.0 | 0.006 |
> > > | Training grad. norm | 1.2 | 0.003 |
> > > | Dormant neuron ratio | 290.1 | 0.580 |
> > >
> > > Local redundancy adds only 1.65% of total training time, despite the Continual ImageNet experiment requiring computations of plasticity metrics on each of 500 tasks. As you pointed out, local redundancy is more expensive than many other simple plasticity metrics, but remains a small fraction of the overall training cost. In practice, since local redundancy essentially requires one more forward and backward pass, the fraction of training cost is approximately equal to the number of batches before computing the local redundancy once; if computing local redundancy only at the end of each epoch or some large number of batches for a checkpoint, it adds very little to total computation. We will include this table in the revision.
> > >
> > > **Ablation of the synthetic data.** We compared three input types: Gaussian noise (independent per pixel), our synthetic shapes, and real unseen images from the next task. The real-image variant acts as a desired baseline but requires knowing the future task and holding out part of that dataset, and is thus not applicable in general. We show the correlation of the local redundancy estimate between each type of synthetic data and this baseline in the table below. We will also include a plot of the local redundancy estimate from each of the three input types, showing that Gaussian noise behaves erratically at various points during training, producing a noisier estimate, whereas the synthetic shapes closely track the real-image baseline. In summary, various synthetic data generation methods produce similar lower bounds, a simple choice of synthetic data suffices to achieve close to the desired real-data baseline, and Theorem 3.2 establishes that any valid inputs functions as a lower bound, so the theoretical guarantees are not broken based on the choice of synthetic data. We will include these ablation details in the appendix.
> > >
> > > | Synthetic input | Pearson \\(r\\) vs. real baseline | \\(R^2\\) vs. real baseline | MSE vs. real baseline |
> > > |---|---|---|---|
> > > | Shapes | 0.964 | 0.929 | 0.067 |
> > > | Gaussian noise | 0.847 | 0.718 | 2.474 |
> > >
> > > **Isolating the effect of the synthetic data.** We agree the estimator can be described as a synthetic gradient sensitivity metric. That is what Theorem 3.4 establishes: the expected squared gradient norm under uniformly randomized targets lower bounds local redundancy, and local redundancy in-turn is a measure of plasticity. The contribution of the formulation is the formal justification for why this specific construction is meaningful. Finally, without the uniform randomization over targets, one gets a different heuristic without the same theoretical guarantee.

---

### Official Review · Reviewer_9MCT · 2026-03-12

**Soundness:** 4
**Presentation:** 3
**Significance:** 4
**Originality:** 3
**Overall Recommendation:** 5
**Confidence:** 3

**Summary:**

This paper introduces local redundancy, an information-theoretic measure of neural network plasticity (in the learning-theoretic sense, not in the neuroscientific sense) grounded in universal coding theory. The core contributions involves two complementary steps: 1) identifying model plasticity with the network's "information radius", i.e. its sensitivity to small parameter changes; and 2) providing a very tractable lower bound using the (squared) gradient on a synthetic dataset. Experiments on continual image classification and time series transfer learning show that local redundancy outperforms existing plasticity metrics, and notably enables checkpoint selection even when validation loss is approximately constant across checkpoints.

**Compliance With Llm Reviewing Policy:**

Affirmed.

**Key Questions For Authors:**

* The measure is defined in terms of distributions reachable by a single gradient step, which fits with the authors' goal of making the measure explicitly local. Learning, however, unfolds over many gradient steps and involves longer-range changes to the loss landscape. Do the authors believe local redundancy captures useful information about these longer-horizon dynamics? For instance, can it distinguish a network that has globally lost plasticity from one that is merely at a temporarily flat region of the loss surface?
* What are the absolute $r$ and $\rho$ values in Table 1, both before and after regressing out task number?

**Limitations:**

Limitations are addressed only very briefly and in passing in the Conclusion. I very strongly encourage the authors to have a more explicit (sub-)section or paragraph meaningfully engaging with the limitations of their work.

**Strengths And Weaknesses:**

**Strengths**
* The theoretical grounding is principled and non-trivial. Localising compression redundancy to gradient neighbourhoods is a clean and well-motivated formalisation of plasticity, and the gradient-based (rather than Euclidean) neighbourhood definition is particularly well-justified.
* The resulting estimator is tractable and very practically appealing, as it relies only on a single backward pass. The trick with the memorisation dataset is clever.
* The checkpoint selection result is an intriguing finding -- since local redundancy continues to vary after validation loss has converged, local redundancy can be used to track learning dynamics where common metrics lose resolution.

**Weaknesses**
* The tightness of the bound is not discussed. I agree with the authors that in the learning theory it is more common to report correlations rather absolute values, this still deserves an explicit discussion. For example, is the gap consistent across architectures or training stages? What does this imply for the plasticity estimate?
* Table 1 reports only correlation improvements over a task-number baseline, not absolute values. Without knowing the raw $r$ and $\rho$, it is much more difficult to judge local redundancy on its own terms.
* The use of a memorisation dataset for the lower bound is clever, but unfortunately is not spelled out in sufficient detail, or in a way that makes intuitive sense for less mathematically-oriented readers. This key idea of the paper would have a wider impact if the authors could explain the relevance of the memorisation task in a more accessible way.

---

> ### Author Rebuttal · Authors · 2026-03-30
>
> Thank you for the very positive and thoughtful review. We are especially glad that you found the gradient-neighborhood definition well motivated and the synthetic-memorization estimator practically appealing.
>
> We agree the paper should discuss bound tightness more explicitly. Our claim is not that the gradient-based estimator is a calibrated absolute estimate of local redundancy. Rather, it is a non-vacuous lower bound whose relative ordering is useful for tracking plasticity. The gap can vary across architectures and training stages because it depends on how well the synthetic inputs saturate the locally reachable capacity. This is exactly why Section 4.3 studies memorization capacity on the synthetic datasets, and why the conclusion frames stronger saturation as future work.
>
> We also agree the memorization construction deserves a more intuitive explanation. The core idea is that synthetic memorization asks how much information about random targets a tiny update can encode. If many random target assignments can be fit after an infinitesimal step, then the nearby model family contains many distinguishable conditional distributions. Sampling labels from $P_{\theta_0}$ is what turns this intuition into the lower bound in Theorem 3.4 via the entropy cancellation in Eq. (11).
>
> Regarding longer-horizon dynamics, local redundancy is explicitly a one-step quantity, so we do not claim it is a complete global theory of plasticity loss. Our motivation is that optimization proceeds through local updates, and empirically the quantity still predicts later task performance and checkpoint quality after validation loss has plateaued. We agree the limitations should be made more explicit on this point.

---

> > ### Author Rebuttal · Reviewer_9MCT · 2026-04-01
> >
> > Thanks for the response. This is very helpful, but the second weakness and second question in my review remain unaddressed. What are the absolute $r$ and $\rho$ values in Table 1, both before and after regressing out task number?

---

> > > ### Author Response · Authors · 2026-04-05
> > >
> > > Thank you for the follow-up! Below we report the absolute correlation values requested:
> > >
> > > | Metric | Pearson \\(r\\) | Spearman \\(\\rho\\) | Pearson \\(r\\) (res.) | Spearman \\(\\rho\\) (res.) |
> > > |---|---|---|---|---|
> > > | Task number | \\(-0.513 \\pm 0.039\\) | \\(-0.495 \\pm 0.039\\) | --- | --- |
> > > | Local redundancy | \\(0.169 \\pm 0.044\\) | \\(0.436 \\pm 0.040\\) | \\(0.084 \\pm 0.045\\) | \\(0.211 \\pm 0.044\\) |
> > > | Distance from init | \\(-0.547 \\pm 0.038\\) | \\(-0.476 \\pm 0.040\\) | \\(-0.053 \\pm 0.045\\) | \\(0.027 \\pm 0.045\\) |
> > > | Weight norm | \\(-0.547 \\pm 0.038\\) | \\(-0.467 \\pm 0.040\\) | \\(-0.057 \\pm 0.045\\) | \\(0.034 \\pm 0.045\\) |
> > > | Dormant neuron ratio | \\(-0.198 \\pm 0.044\\) | \\(-0.405 \\pm 0.041\\) | \\(0.097 \\pm 0.045\\) | \\(0.080 \\pm 0.045\\) |
> > > | Training grad. norm | \\(0.086 \\pm 0.045\\) | \\(0.385 \\pm 0.041\\) | \\(0.018 \\pm 0.045\\) | \\(0.208 \\pm 0.044\\) |
> > >
> > > Continual ImageNet performance shows a strong linear decrease over time, and this accounts for most of the signal. Thus, computing raw correlations essentially reduces to correlation with task number, which are very large for some predictors, as shown in the second table below. Local redundancy has a lower correlation with task number, whereas predictors like dist from init and weight norm simply increase almost linearly with training, and thus have a larger raw correlation; no predictor's raw correlation is larger than the task number's with statistical significance. For this reason, we use the correlation between a predictor and the residual after regressing out the baseline effect of task number. We believe this quantity is also more intuitive, since better plasticity measures should correlate stronger with the residual performance, rather than simply track the task number. We want to clarify that we are not subtracting correlations, since that procedure is somewhat unusual and the result is hard to interpret; we will revise Table 1's caption to make this much more explicit, since the current wording (such as "correlation improvement") is ambiguous.
> > >
> > > | Metric | Pearson \\(r\\) with task # | Spearman \\(\\rho\\) with task # |
> > > |---|---|---|
> > > | Distance from init | \\(0.979\\) | \\(0.993\\) |
> > > | Weight norm | \\(0.973\\) | \\(0.988\\) |
> > > | Dormant neuron ratio | \\(0.540\\) | \\(0.884\\) |
> > > | Local redundancy | \\(-0.182\\) | \\(-0.523\\) |
> > > | Training grad. norm | \\(-0.128\\) | \\(-0.407\\) |

---

### Official Review · Reviewer_GRg6 · 2026-03-13

**Soundness:** 2
**Presentation:** 2
**Significance:** 3
**Originality:** 3
**Overall Recommendation:** 4
**Confidence:** 3

**Summary:**

The paper introduces local redundancy, an information-theoretic measure intended to quantify the plasticity of neural networks, i.e., their ability to adapt to new tasks in continual and transfer learning. The authors derive the concept from universal compression theory and define local redundancy as the worst-case redundancy of a local model family around the current parameters along gradient directions. Because computing this quantity exactly is intractable, the paper proves that the expected squared gradient norm on a synthetic memorization task provides a computable lower bound. Empirical results on continual image classification and time-series transfer learning suggest that this measure correlates with downstream task performance better than several existing plasticity metrics and can help select pretraining checkpoints when validation loss has plateaued.

**Compliance With Llm Reviewing Policy:**

Affirmed.

**Final Justification:**

The rebuttal has addressed my concerns.

**Key Questions For Authors:**

1. The paper proposes tracking plasticity loss and predicting future task accuracy through the proposed local redundancy metric. However, it is not entirely clear what practical actions this prediction enables. For instance, if the predicted performance on future tasks is low, what concrete interventions can be taken during training to improve plasticity? Clarifying the practical use cases of this metric would strengthen the motivation.

2. The meaning of *local* in phrases such as “plasticity is inherently local” and “local redundancy” is somewhat unclear. The term appears different from the notion of locality used in optimization (e.g., local minima or local curvature). It would be helpful to provide a precise definition and explanation of what locality refers to in this context.

3. Related to the previous point, the second sentence of Section 3.2 states that the metric “measures what a network can learn from its current state, not what the architecture could learn in principle.” It is unclear what exactly is meant by the network’s *state*. Does this simply refer to the current parameter values? If so, the statement that a network “learns from its own parameters” is somewhat confusing and could benefit from clarification.

4. Section 3.4 presents several bounds, but their implications are not clearly explained. It would be helpful if the authors elaborated on what insights these bounds provide and how they contribute to understanding plasticity or guiding practical model training.

**Limitations:**

yes

**Strengths And Weaknesses:**

### Strengths

- The paper includes both theoretical analysis and empirical experiments.
- Using information-theoretic ideas to study neural network learning dynamics is a reasonable and interesting direction.

### Weaknesses

- The main theoretical results appear relatively lightweight, with the proofs of the key theorems consisting of only a few lines. As a result, the technical depth of the theoretical contribution seems limited.
- Several important terms and claims are presented in a rather vague way. In particular, notions such as *local*, *global*, and *memorization* are used repeatedly but are not clearly or rigorously defined, which makes parts of the argument difficult to follow.

---

> ### Author Rebuttal · Authors · 2026-03-30
>
> Thank you for the thoughtful review! We agree that some parts of Section 3 should be stated more explicitly. Our main claim is narrower than the review may suggest: the paper aims to define and measure plasticity in a principled way, not to propose a new intervention for preserving plasticity. We also think the main theoretical contribution is more than a short proof sketch. The key step is defining the right local information-theoretic quantity and then showing that it reduces to a computable first-order estimator.
>
> Q1: What practical action does this metric enable?
> Our goal in this paper is measurement of plasticity, not intervention. Nevertheless, we do show one practical use case in checkpoint selection: in Table 2, local redundancy can select pretraining checkpoints better than existing metrics like using the lowest pretraining validation loss. More broadly, we view it as a state variable for long training pipelines: it can be used to monitor loss of forward adaptability over time, and to decide when existing interventions such as replay, reset, or regularization may be worth applying. We agree the paper should state the scope and use cases more directly.
>
> Q2: What does “local” mean here?
> Here, “local” refers to the local model family defined in Section 3.2, not to locality in the sense of local minima or local curvature. Specifically, it means the family of predictors reachable from the current parameter vector $\theta_0$ by one gradient step of size at most $\epsilon$ on an arbitrary size-$n$ dataset. The intended contrast is between this local family and the full architecture-level model family. We will revise the wording so this distinction is made explicit before we use the term “local” more broadly.
>
> Q3: What is meant by the network’s “current state”?
> Yes, this simply means the current parameter values, and therefore the current input-output map. The intended point was that local redundancy measures what is reachable from the model’s present parameters, rather than from random initialization and after many optimization steps, but this may have caused more confusion than clarification; we agree the current phrasing can be made clearer and will revise it to make it clearer!
>
> Q4: What do the bounds in Section 3.4 actually imply?
> These bounds are the reason the memorization experiment is meaningful. Theorem 3.2 shows that memorizing randomized targets gives a lower bound on worst-case redundancy, for any choice of inputs. Proposition 3.3 shows that for the local model family, the maximizing parameter is obtained to first order by stepping in the corresponding gradient direction. Theorem 3.4 then gives a first-order lower bound of the form $R_n^{**}(\theta_0,\epsilon)\ge \epsilon,\mathbb{E}|\nabla_\theta L|_2^2 + O(\epsilon^2)$. This is exactly the link from the abstract local redundancy definition to the estimator used in the experiments. We agree this role should be explained more clearly in Section 3.4; we will add a reference to the intuition figure in Figure 1, which will help clarify things.
>
> Thank you again for the helpful comments! We think the main issues you raise are about exposition and scope, and we will revise the paper to make the definitions, intended claim, and practical role of the metric clearer.

---

> > ### Author Rebuttal · Reviewer_GRg6 · 2026-04-04
> >
> > Thanks for the response.

---

### Official Review · Reviewer_ZxkT · 2026-03-14

**Soundness:** 3
**Presentation:** 3
**Significance:** 2
**Originality:** 3
**Overall Recommendation:** 4
**Confidence:** 3

**Summary:**

The authors propose local redundancy as a theoretically principled, analytically tractable and empirically predictive measure of plasticity in neural networks. They motivate local redundancy as an information-theoretic geometric interpretation of distributional plasticity, prove that it’s lower bounded by synthetic memorization and calculable via gradients. Finally, they demonstrate empirically that it outperforms existic plasticity proxies for both correlation with future task accuracy and unsupervised checkpoint selection.

**Compliance With Llm Reviewing Policy:**

Affirmed.

**Final Justification:**

This is promising work; I'm maintaining my score due to the limited analysis of local redundancy with stability.

**Key Questions For Authors:**

* **Q1:** Can you characterize what happens to local redundancy at catastrophic forgetting transitions? The central challenge in continual learning is forgetting, and without evidence that local redundancy interacts meaningfully with forgetting dynamics, the practical applicability of the measure remains unclear.
* **Q2:** Can you provide additional experimental details? Absolute correlation values (not just improvement over baseline) for Table 1; standard deviations of local redundancy estimates across synthetic batches; sensitivity of results to the number of synthetic samples n, etc.
* **Q3:** How would you design an experiment establishing local redundancy as causally related to plasticity, rather than merely correlated? The geometric interpretation is suggestive, and if validated causally it would powerfully connect information theory, mechanistic interpretability, and continual learning.
* **Q4:** What are the primary obstacles to directly characterizing average-case local redundancy? The geometric "information radius" interpretation is elegant and the apparent looseness of the bounds may stem from the worst-to-average-case asymptotic equivalence not holding at locality.

**Limitations:**

The authors acknowledge the small model scale (≤5.5M parameters) and the gap between achieved memorization capacity (~1–2 bits/param) and the theoretical ceiling (16 bits/param). They do not discuss the absence of forgetting analysis, interaction with continual learning methods, or the causal status of the correlation results.

**Strengths And Weaknesses:**

**S1 (Predictive power):** Local redundancy achieves the highest correlation improvement with future task accuracy among all metrics tested (Table 1).
* **S2 (Unsupervised checkpoint selection):** Selects better pretraining checkpoints than lowest validation loss *without downstream labels* — a property no other plasticity metric achieves (Table 2).
* **S3 (Efficient tractability):** The chain from Shtarkov sum → Jensen bound → local MLE → gradient norm is clean, reducing an intractable quantity to a single backward pass on synthetic noise.
* **W1 (No catastrophic forgetting analysis):** The experiments measure only forward transfer. The central tension in continual learning is the stability-plasticity tradeoff, yet no backward transfer metrics are reported.
* **W2 (Asymptotic appeal):** The geometric "information radius" interpretation comes from average-case redundancy (Eq. 5), but the analysis is on worst-case redundancy (Eq. 1). These are asymptotically equivalent in samples n, but the local family $\Theta(\theta_0, \epsilon)$ is infinitesimal (and implicitly depends on n) — it is unclear the asymptotic regime applies.
* **W3 (No interaction with interventions):** The measure is evaluated in isolation. How does local redundancy behave under EWC, replay, shrink-and-perturb, or weight decay? A measure of plasticity should respond predictably to known plasticity-preserving interventions.
* **W4 (Correlational, not causal):** The results show local redundancy *correlates* with future performance, but the lower bound does not establish causation. It remains possible that local redundancy and plasticity are joint effects of a common latent cause (e.g., distance from initialization, which achieves 7.3%).
* **W5 (Statistical reporting):** Figure 4 uses ±1 standard deviation shading, which covers ~68% of seeds but is already quite wide; would 95% be statistically significant?

---

> ### Author Rebuttal · Authors · 2026-03-30
>
> Thank you for the careful and constructive review. We are glad the two main intended contributions came through: a principled local complexity measure of plasticity, and a tractable one-backward-pass estimator via synthetic memorization.
>
> We agree that the current experiments study forward plasticity rather than the full stability-plasticity tradeoff. In continual ImageNet, this is also why Table 1 reports improvement after regressing out task number: task index is already highly predictive because performance drifts downward over training, so our aim was to isolate signal beyond that global trend. We agree an explicit forgetting or backward-transfer analysis would strengthen the paper, and we should state that limitation more clearly.
>
> Your point about worst-case versus average-case redundancy is also well taken. Our formal results are on worst-case local redundancy, since the Shtarkov characterization is what yields Theorems 3.2 and 3.4. The information-radius discussion is meant as geometric motivation only. We do not claim a finite-sample equivalence for the local family, and we should make that distinction clearer in the presentation.
>
> We also agree the present paper establishes predictive utility, not causation. A natural next step would be to intervene on plasticity directly, for example via replay, EWC, shrink-and-perturb, or regularization, and test whether induced changes in local redundancy systematically precede changes in later adaptation. We view that as important future work rather than a current claim.
>
> Finally, Figure 4's $\pm 1$ std band is descriptive only. The main inferential comparison is Table 2, which reports mean $\pm$ stderr over 120 seeds. We also agree the limitations section should be more explicit overall.

---

> > ### Author Rebuttal · Reviewer_ZxkT · 2026-04-03
> >
> > I appreciate the intellectual honesty. Would it be possible to do some additional analysis of stability for this cycle? I believe that would significantly strengthen the paper.

---

> > > ### Author Response · Authors · 2026-04-05
> > >
> > > Thank you for the suggestion! We think this is a really good idea and helps round out the Continual ImageNet experiment. Specifically, we analyze local redundancy and the stability-plasticity tradeoff: we compute correlations between each plasticity metric and degradation in past task performance (specifically, the original accuracy on a previous task minus the new accuracy after training on the next task, and regressing out the task number as before; positive values indicate more forgetting):
> > >
> > > | Metric | Pearson \\(r\\) (res.) | Spearman \\(\\rho\\) (res.) |
> > > |---|---|---|
> > > | Local redundancy | \\(0.130 \\pm 0.045\\) | \\(0.235 \\pm 0.044\\) |
> > > | Distance from init | \\(-0.020 \\pm 0.045\\) | \\(0.037 \\pm 0.045\\) |
> > > | Weight norm | \\(0.022 \\pm 0.045\\) | \\(0.041 \\pm 0.045\\) |
> > > | Dormant neuron ratio | \\(-0.053 \\pm 0.045\\) | \\(-0.029 \\pm 0.045\\) |
> > > | Training grad. norm | \\(0.062 \\pm 0.045\\) | \\(0.236 \\pm 0.044\\) |
> > >
> > > Local redundancy indeed shows a positive correlation with forgetting (higher local redundancy associated with more forgetting), and has among the largest correlations with forgetting of all the plasticity metrics tested, further validating the theoretical grounding behind local redundancy. We will include this table and related explanations in Section 4 to strengthen the existing experiments.

---

### Decision · Program_Chairs · 2026-04-30

**Decision:**

Accept (spotlight)

**Comment:**

The paper proposes local redundancy, a differentiable measure of neural network plasticity grounded in information-theoretic geometry. While it has a theoretical lower bound tied to synthetic memorisation, in practice it turns out to be a strong predictor of future task performance. Empirically, it outperforms existing proxy metrics and can be used to select model checkpoints without relying on labelled data.

This paper sees substantial agreement on the strengths and limitations. The key strengths identified are: the elegant theoretical grounding, the tractability of the single-backward-pass estimator, and the clarity of the analysis; while the concerns were mostly about: the statistical reporting of the results, the computational costs of the measure, and then concerns about scope (i.e., questions about catastrophic forgetting, limited model coverage, and possible interventions relying on the result).
The authors engaged thoroughly during the discussion period and successfully addressed the concerns. In particular, they provided new results (such as the absolute correlation values, stability-plasticity correlations, and computational cost breakdowns), new directions, pointing to the relevant sections, or just clarifying their scope.
After the discussion period, there is a general consensus towards acceptance, and **I also support acceptance of the paper.**